# The Role of the MntABC Transporter System in the Oxidative Stress Resistance of *Deinococcus radiodurans*

**DOI:** 10.3390/ijms26199407

**Published:** 2025-09-26

**Authors:** Binqiang Wang, Renjiang Pang, Chunhui Cai, Zichun Tan, Shang Dai, Bing Tian, Liangyan Wang

**Affiliations:** 1Institute of Biophysics, College of Life Sciences, Zhejiang University, Hangzhou 310058, China; wangbinqiang@zju.edu.cn (B.W.);; 2State Key Laboratory of Clean Energy Utilization, Institute of Carbon Neutrality, Zhejiang University, Hangzhou 310027, China; 3College of Life Sciences, Nanjing Agricultural University, Nanjing 210095, China

**Keywords:** *Deinococcus radiodurans*, MntABC, Mn ion transport, oxidative stress-resistance

## Abstract

The accumulation of high levels of manganese ions complexed with small molecules has been proposed as a pivotal factor contributing to the extraordinary radiation resistance of *Deinococcus radiodurans*. However, the molecular mechanisms governing the manganese ion homeostasis remain elusive. In this study, we characterize the role of the MntABC transporter system for Mn ion accumulation in *D*. *radiodurans*. Its cellular membrane localization is unequivocally demonstrated through fluorescence labeling techniques. Mutation of the protein components of the MntABC led to a significant decrease in intracellular Mn ion accumulation, concomitant with impaired cellular growth, decreased resistance against hydrogen peroxide, and gamma-ray irradiation-induced oxidative stresses, indicating that the MntABC system plays an indispensable role in resistance of *D*. *radiodurans* to oxidative stresses. Protein structure modeling and molecular docking are employed to analyze the key active sites of the MntABC proteins and their intermolecular interactions. The results demonstrate that the MntABC system is essential for maintaining Mn ion homeostasis and the oxidative stress resistance of *D. radiodurans*.

## 1. Introduction

*Deinococcus radiodurans* represents an extremophilic bacterium known for its exceptional resistance to multiple environmental stresses, including ionizing radiation (IR), ultraviolet (UV) radiation, oxidative stress, and desiccation [1,2,3]. This bacterium has been extensively employed as a model organism for investigating the mechanisms underlying oxidative stress resistance. *D. radiodurans* exhibited a distinctive proteome protection against oxidative damage due to its remarkable ROS scavenging capacity relative to other bacteria [4,5], which is crucial for maintaining protein functionality during stress conditions and facilitating efficient DNA damage repair processes [6,7,8]. This remarkable ROS scavenging capability is thought to be mediated through synergistic actions of both enzymatic components (e.g., catalases, superoxide dismutases, and peroxidases) and non-enzymatic antioxidants including divalent manganese (Mn) ion complexes [5,6,9,10,11,12,13].

Among the essential metal elements, the Mn ion serves as a component in cellular antioxidant defense systems [10,12,14]. Interestingly, it is found that the Mn/Fe ratio of most IR-resistant bacteria is higher than that of IR-sensitive bacteria [14,15,16]. In the *D. radiodurans*, intracellular Mn^2+^ concentrations are maintained within the range of 0.2–4 mM, with Mn/Fe ratios reaching 0.24, representing a 5–10 fold higher level than in the IR-sensitive *Escherichia coli* [7]. This pronounced Mn^2+^ accumulation has been mechanistically linked to the organism’s extraordinary tolerance to ionizing radiation and oxidative stress [14]. At the molecular level, Mn^2+^ does not directly participate in redox reactions with ROS, but it works in the complex form by coordination with low molecular weight metabolites—including phosphate groups, amino acid residues, oligopeptides, or nucleotide derivatives—generating small-molecule Mn-antioxidant complexes. These Mn-antioxidant complexes exhibit ROS-scavenging capacity of superoxide radicals (O_2_•^−^), hydroxyl radicals (•OH), and hydrogen peroxide (H_2_O_2_) [17,18]. Moreover, Mn^2+^ functions as an essential cofactor for many proteins of *D. radiodurans*, such as manganese superoxide dismutase (MnSOD), where it coordinates within the enzyme’s active site to catalyze the disproportionation of superoxide radicals [19]. Consequently, maintaining intracellular manganese ion homeostasis is crucial for the formation of the Mn-antioxidant complexes, thereby contributing to the survival of *D. radiodurans* under extreme environments.

The intracellular manganese homeostasis in prokaryotes is regulated by two functionally distinct classes of transporters: uptake system and efflux system. The manganese acquisition in prokaryotes primarily occurs through two key proteins: the proton-coupled Nramp family transporter MntH, and the ATP-binding cassette (ABC) transporter complex MntABC, which is alternatively named SitABCD in some bacterial species [20,21]. Conversely, manganese extrusion is principally mediated by the cation diffusion facilitator (CDF) family transporter MntE. In the extremophile *D. radiodurans*, genetic studies have revealed that *dr1709* encoded a MntH transporter; when it was disrupted, the mutant demonstrated radiation-sensitive phenotypes and decreased intracellular Mn contents [22,23]. *dr1236* encodes a MntE homolog, and its disruption confers enhanced resistance of *D. radiodurans* cells to H_2_O_2_, UV, and γ-radiation but increased susceptibility to manganese toxicity [24]. Notably, the genomic organization of ABC-type manganese transporters in *D. radiodurans* exhibits atypical characteristics. While canonical MntABC systems form a contiguous operon encoding three core components (periplasmic binding protein MntC, transmembrane permease MntB, and ATPase subunit MntA) [20], the predicted manganese ABC transporter in the *D. radiodurans* genome displays a non-canonical gene distribution: ATPase MntA (DR2284), permease MntB (DR2283), and periplasmic-binding protein MntC (DR2523), which are genomically dispersed rather than organized in an intact operon [10]. Although the MntABC transporter has been predicted, the exact physiological function of the MntABC transporter system in *D. radiodurans* is still not fully understood. Previously, a single-gene-disrupted mutant of DR2523 (DrMntC) exhibited little different phenotype compared with the wild-type strain after being treated with H_2_O_2_ and UV light [22], and the disruption of DR2523 alone could not remarkably change the intracellular manganese level in *D. radiodurans* [23]. The lack of comprehensive study of the MntABC operon hindered understanding the structure–function relationship of the distinctive manganese transport system, especially concerning its possible roles in maintaining Mn ion homeostasis and oxidative stress resistance of *D*. *radiodurans*.

In this study, we characterized the MntABC manganese transport system in *D. radiodurans* and demonstrated its essential role in regulating intracellular manganese homeostasis and involvement in oxidative stress resistance. Structural analysis revealed that the key catalytic residues of MntABC proteins and their specific intermolecular interactions might be critically involved in Mn ion transport. These findings provide insights into the molecular mechanism by which the MntABC system mediates manganese ion homeostasis and oxidative stress resistance in *D*. *radiodurans*.

## 2. Results

### 2.1. Identification of the MntABC System in D. radiodurans

Through sequence alignment analysis, the amino acid sequences of DR2523, DR2283, and DR2284 with their respective homologous proteins from *Escherichia coli*, *Staphylococcus aureus*, *Paracoccus denitrificans*, and *Streptococcus pneumoniae* were compared. The results demonstrated that all three proteins maintain substantial evolutionary conservation. R2523 displays 32% sequence identity to that of *E. coli* MntC, DR2283 exhibits 40% sequence identity, and DR2284 displays 41% sequence identity to their respective homolog proteins in *E. coli* (Appendix A). To experimentally verify whether *dr2283* and *dr2284* are co-transcribed as a single operon, we conducted PCR-based validation targeting the intergenic region between these two genes (Appendix A). The result established that *dr2283* and *dr2284* are indeed organized as a single transcriptional unit and undergo coordinated expression (Appendix A). Subsequently, gene knockout mutants of *dr2523* (*drmntC*), *dr2283* (*drmntB*), and *dr2284* (*drmntA*), along with a combinatorial mutant deficient in all three genes (designated *Δdr2523*, *Δdr2283*, Δ*dr2284*, and *ΔdrMntABC*, respectively), were successfully constructed and verified (Appendix A).

Next, we investigated the cellular localization of protein homologs of *D. radiodurans* MntABC system using fluorescent labeling. In the classical bacterial MntABC system, the metal substrate-binding protein MntC is localized in the periplasmic space and anchored to the cell membrane. The permease MntB is embedded across the membrane, while the ATP-binding protein MntA is predominantly located in the cytoplasm. Utilizing pRADG vectors containing the *eGFP* gene, we constructed target gene-*eGFP* fusion expressing strains: *dr2283*-*eGFP*, *dr2284*-*eGFP*, *dr2523*-*eGFP*, and a control strain transformed with empty pRADG vector. Fluorescence microscopy showed that in the wild-type R1 control group, the expressed eGFP, was dispersed in the cytoplasm (Figure 1A); however, the transformants expressing DR2283-eGFP, DR2284-eGFP, and DR2523-eGFP exhibited green fluorescence located in cell membrane region. Specifically, the DR2283-eGFP showed higher fluorescence intensity at the junctions of cell membranes within each tetrad cell, which is consistent with the fact that DR2283 possesses multiple transmembrane domains and α-helices. Simultaneously, DR2523-eGFP also displayed high green fluorescence at the cell membrane region, indicating that DR2523 as a MntC homolog is located on the cell membrane and may partially embed into it for metal ion binding during uptake. The fluorescence intensity of DR2284-eGFP in the membrane region was weaker, possibly due to the absence of transmembrane domains in DR2284. Most parts of the MntA homolog are cytoplasmic and associate with MntB and MntC via protein–complex interactions [25].

Quantitative analysis of metal ion content in mutant cells cultured in TGY (tryptone, glucose, and yeast extract) was conducted using ICP-MS, with wild-type cells serving as controls. As shown in Figure 1B, analysis of intracellular Mn levels revealed a significant decrease (>50%) in the *ΔdrMntABC* mutant strain compared to the wild type, indicating that deletion of the *mntABC* system impairs the cell’s capacity for Mn uptake. Each of the single mutants of *dr2523*, *dr2283*, and *dr2284* exhibited less decrease in intracellular Mn ions than the *ΔdrMntABC* mutant. These findings suggested that *dr2523* (DrMntC), *dr2283* (DrMntB), and *dr2284* (DrMntA) jointly form the MntABC transporter in *D. radiodurans*. On the other hand, the intracellular levels of Fe, Zn, and Cu ions in the mutants showed no significant differences compared to the wild-type strain. To investigate potential ion transport specificity of the MntABC, we performed metal ion sensitivity assays of the mutant strains, comparing them with the wild-type strain (Figure 1C). The mutations led to reduced sensitivity to Mn ions (Mn^2+^) (Appendix A). However, the increased zinc ion sensitivity was observed for both the MntABC mutant and the *dr2523* mutant, implying that the MntABC system may also participate in the efflux process or homeostasis of other metal ions.

### 2.2. Mutation Deficiency of the MntABC System Affects Cellular Growth

The MntABC system plays an essential role in maintaining cellular metal ion homeostasis, with its dysfunction resulting in impaired bacterial growth. To characterize the physiological role of the MntABC system in *D*. *radiodurans*, we performed comparative growth analyses between wild-type strain R1 and various mutant strains. As shown in Figure 2A, deletion of any component within the MntABC system caused significant growth impairment. The *Δdr2284* and *Δdr2283* mutants displayed nearly identical growth phenotypes, consistent with their co-localization within a single operon and suggesting functional interdependence between DR2284 (DrMntA) and DR2283 (DrMntB). Notably, both *Δdr2523* (DrMntC) and *ΔdrMntABC* strains exhibited more growth defects (Figure 2A). This pronounced growth inhibition is likely due to a blockage in metal uptake, given that the periplasmic DR2523 (MntC) protein contains essential metal-binding domains, underscoring its critical functional contribution to the MntABC transporter system. To further validate gene-specific function, we performed gene complementation experiments using mutant strains. As presented in Figure 2B, individual gene complementation in single-deletion mutants or *ΔdrMntABC* restored growth to the level of wild-type (R1). However, partial complementation of the *ΔdrMntABC* strain revealed that DR2284 (MntA) complementation provided minimal growth restoration, whereas complementation of DR2523 (MntC) or DR2283 (MntB) facilitated partial recovery of *ΔdrMntABC*. These findings suggest that the MntABC transporter system is fundamentally required for normal growth of *D*. *radiodurans* through its role in maintaining manganese ion homeostasis.

### 2.3. MntABC Is Involved in the Resistance of D. radiodurans to Oxidative Stress

Using real-time quantitative PCR, we confirmed that all three *mntABC* genes were upregulated in the R1 strain under oxidative stress conditions triggered by hydrogen peroxide and ionizing radiation (Appendix A). To evaluate the contribution of the MntABC system to oxidative stress resistance of *D*. *radiodurans*, we conducted analyses of the stress tolerance between mutant strains (*Δdr2283*, *Δdr2284*, *Δdr2523*) and wild-type R1. All mutants displayed dose-dependent sensitivity and expression to ionizing radiation (Figure 3A) and exhibited pronounced sensitivity to H_2_O_2_ exposure (Figure 3B). Intracellular ROS quantification revealed that following treatment with 60 mM H_2_O_2_, the *Δdr2523*, and *ΔdrMntABC* mutant strains accumulated approximately 1.5-fold higher ROS levels compared to wild-type controls (Figure 3C). The ROS levels in both *Δdr2283* and *Δdr2284* mutants were also significantly elevated relative to the wild-type strain (Figure 3C). We propose that the MntABC deficiency blocked Mn ion uptake, thereby compromising cellular capacity to employ manganese-dependent antioxidants for ROS scavenging and protein protection under oxidative stress. Our result is different from that of a previous study on the mutant of DR2523 (DrMntC), which exhibited little phenotypic difference compared with the wild-type strain after being treated with H_2_O_2_ [22]. This discrepancy can be explained by the different mutant construction methods in our study and the previous study. In our study, the mutant is constructed by replacing the whole targeted gene with antibiotic gene through homologous recombination, while in the previous study an insertion mutation of an antibiotic gene into the targeted gene was used [22].

### 2.4. Key Sites Within MntABC Proteins Involved in Mn Ion Transport and Oxidative Stress Resistance

By combining AlphaFold2-based in silico protein structure prediction with molecular docking simulations, we modeled the interactions of ATP/Mg^2+^ with MntA’s ATPase domain, as well as Mn^2+^ with both the transmembrane MntB and periplasmic protein MntC (Figure 4A–C). These computational approaches allowed us to predict the key substrate-binding residues across all three protein components. In the DrMntA ATPase domain, residues S41 and D157 were computationally predicted as essential for the Mg^2+^/ATP coordination (Figure 4A). Subsequent intracellular Mn ion assay and survival phenotype analysis of the corresponding gene complementation strains with site-directed mutations at these positions were performed. The site-directed mutations of S41, D157, and both sites led to decreased intracellular Mn ion levels and reduced H_2_O_2_ stress resistance (Figure 4D,G). For the DrMntB, a conserved acidic pocket formed by H48 and D44 was predicted to serve as the primary Mn^2+^ chelation site (Figure 4B). The DrMntC possesses a tetrahedral Mn^2+^ coordination sphere involving H61, H123, H189, and D266 (Figure 4C). Analysis of corresponding gene complementation strains with site-directed mutations of these residues in both DrMntB and DrMntC indicated that the site-directed mutations resulted in decreased intracellular Mn accumulation (Figure 4E,F) and oxidative stress tolerance (Figure 4H,I). These findings suggest that the predicted sites in MntABC proteins play crucial roles in Mn ion transport and oxidative stress resistance.

### 2.5. Interactions Between the Subunits of the MntABC System Impact on Manganese ion Transport

The MntABC system represents an ATP-binding cassette (ABC) transporter complex dedicated to manganese ion translocation, functioning through coordinated subunit interactions to fulfill transmembrane transport [26]. Utilizing AlphaFold 2, we computationally predicted the structural architecture of the MntABC complex of *D*. *radiodurans*, which comprises a single DrMntC monomer, an DrMntB dimer, and an DrMntA dimer (Figure 5A). Protein–protein interaction analysis identified the residues R15, D134, and D244 from DrMntB forming hydrogen bonds and salt bridges with the R246, S253, and R220 of DrMntC, respectively (Figure 5A). Meanwhile, residues D162 and E165 from DrMntB form salt bridges with R145 and R149 of DrMntA (Figure 5A). Additionally, E254 on DrMntC interacts with the backbone of a residue on DrMntB, and K154 of DrMntB interacts with the backbone of a residue of DrMntA (Figure 5A). DrMntB’s intermediary role in bridging DrMntA and DrMntC.

The potential interacting residues were selected for site-directed mutagenesis. To elucidate the impact of site mutations in DrMntB on its interactions with DrMntA and DrMntC, yeast two-hybrid assays were performed. Each mutant of DrMntB was fused to either the DNA-binding domain (BD) or activation domain (AD), while DrMntA and DrMntC were expressed with the reciprocal domains. The results showed that the R15, D134, and D244 substitutions with alanine in DrMntB markedly attenuated yeast colony growth (Figure 5B), indicating that these residues are involved in the interaction with DrMntC. The triple mutant of DrMntB (ITM) abolished almost all the yeast colony growth (Figure 5B), suggesting that R15, D134, and D244 collectively determine the interaction of DrMntB with DrMntC. Analogously, mutations of D162 and E165 in DrMntB substantially impaired the yeast growth phenotypes (Figure 5C), while the double mutant (IDM) led to little yeast colony growth (Figure 5C), confirming the essentiality of D162 and E165 for the interaction of DrMntB with DrMntA.

Furthermore, we complemented the DrMntB triple mutant (ITM), double mutant (IDM), and quintuple mutant (ITM/IDM) gene into the DrMntB mutant (*Δdr2283*) strain to investigate their effect on Mn ion transport. The results revealed that the intracellular Mn^2+^ concentrations in the complemented strains failed to restore to the wild-type level (Figure 5D). Notably, the quintuple mutation (ITM/IDM) decreased the intracellular Mn^2+^ concentration to approximately 31% of that in the wild-type strain (Figure 5D), indicating that these residues of DrMntB are involved in interactions with DrMntA and DrMntC which are important for Mn ion transport. Moreover, the ITM, IDM, and ITM/IDM mutations resulted in a substantial decrease in the mutant cell survival under H_2_O_2_ stress (Figure 5E). Taken together, these findings suggested that DrMntB may interact with DrMntA and DrMntC through the identified sites, which is essential for maintaining intracellular Mn ion homeostasis that confers resistance of *D. radiodurans* against oxidative stress.

## 3. Discussion

In this work, we characterized the MntABC transporter system of extremophile *D*. *radiodurans* with exceptional tolerance to oxidative stress. Our findings revealed that the MntABC system is indispensable for both cellular growth and oxidative stress resistance through its maintaining Mn ion homeostasis. The DrMntB may modulate the transporter’s functionality via specific molecular interactions with DrMntA and DrMntC subunits. 

Mutation of the MntABC system resulted in a significant decrease in intracellular Mn ion concentration, indicating its role in Mn ion transport. We also found that the DrMntC mutant (*Δdr2523*) exhibited pronounced zinc sensitivity, implying that the MntABC may also participate in the efflux process or homeostasis of other metal ions such as Zn ions. These observations are consistent with previous reports of zinc ion sensitivity in MntC mutant strains and Zn binding capacity in MntC homologs [25,27]. A single disruption of DR2523 resulted in the accumulation of Fe ions in *D. radiodurans* [23]. The possible transport and/or efflux function of MntABC on other metal ions in *D. radiodurans* deserves further investigation. Manganese ions constitute indispensable micronutrients supporting cellular proliferation in diverse bacterial species. In the extremophile *D*. *radiodurans*, exogenous Mn^2+^ supplementation induces a distinctive manganese-triggered cell division response [28]. Our evidence demonstrates that the MntABC transporter system is fundamentally required for sustaining cell growth. 

Maintenance of a high intracellular manganese-to-iron (Mn/Fe) ratio constitutes a fundamental determinant of stress resistance in the *D*. *radiodurans* [8,14]. The radiation-resistant bacterial species maintain significantly higher Mn/Fe ratios than their radiation-sensitive counterparts, underscoring the pivotal role of Mn ion homeostasis in their extremophilic robustness [29]. Our previous study identified and characterized the MntE as a functional manganese ion efflux transporter in *D. radiodurans*, and phenotypic analyses revealed that *mntE* knockout mutants exhibit enhanced resistance against hydrogen peroxide, ultraviolet radiation, and γ-ray irradiation [24]. Thus, the Mn ion homeostasis in *D*. *radiodurans* should be controlled by a complicated pathway with multiple genes, including the Mn ion transporter MntABC and the reported MntH [16,22,23], as well as the efflux transporter MntE. Moreover, the Mn ion transporters can be transcriptionally regulated by multiple regulatory proteins such as MntR (DR2539) and Fur homolog (DR0865). It was found that the MntR can binds specifically to the promoter of MntH gene, acting as a transcriptional repressor, while the Fur homolog can bind to the promoters of putative MntABC genes [16,30,31]. However, the regulatory mechanism by which the Mn ion transporters facilitate manganese uptake to achieve a high intracellular Mn/Fe ratio in *D. radiodurans* requires further investigation through combination mutant construction and analysis of the Mn ion transporters and regulators. 

The structure of the MntABC complex in *D. radiodurans* has not yet been elucidated. By integrating AlphaFold 2-based structural predictions with experimental validation through site-directed mutagenesis, we investigated the key sites in the MntABC transporter complex that are involved in Mn ion transport. The structures of MntC homologs, such as PsaA from *Streptococcus pneumoniae,* have been elucidated [25,27]. PsaA binds both Zn^2+^ and Mn^2+^ ions through an identical tetrahedral coordination geometry mediated by four evolutionarily conserved residues [27]. In the predicted DrMntC-Mn^2+^complex, residues H61, H123, H189, and D266 chelate Mn^2+^ to establish a tetrahedral coordination architecture, with these binding sites facilitating Mn^2+^ ion sequestration (Figure 4C). MntA, functioning as the canonical ATPase component of the ABC transporter system, generates energy via Mg^2+^-dependent ATP binding and hydrolysis [32]. The predicted MntA-ATP-Mg^2+^complex exhibits a ligand-binding mode analogous to previously resolved crystal structures of homologous complexes. The HD domain (encompassing residues H48 and D44) in MntB, which coordinates Mn^2+^ ions, constitutes a conserved structural motif characteristic of ABC transporters [33]. Also, we identified the key interaction sites among the three proteins. These protein-protein interfaces of MntABC are indispensable for manganese translocation, which is consistent with other ABC transporters, such as heme-translocating systems [34]. Further mechanistic investigations will leverage cryo-electron microscopy with molecular dynamic simulations to determine the definitive structural and dynamic framework of the MntABC complex.

## 4. Material and Methods

### 4.1. Bacterial Strains and Growth Media

All strains and plasmids utilized in this study are listed in Appendix A. *E*. *coli* strains were cultured aerobically at 37 °C in Luria–Bertani (LB) broth (1% tryptone, 0.5% yeast extract, 1% NaCl) or on LB agar plates (1.2% Bacto-agar). *Deinococcus radiodurans* strains were propagated in TGY medium (0.5% tryptone, 0.1% glucose, 0.3% yeast extract) under aerobic conditions at 30 °C, with solid cultures prepared using 1.5% Bacto Agar. All growth media formulations and antibiotic selection conditions adhered to standardized microbiological protocols.

### 4.2. Bioinformatics and Structural Analysis

Previously characterized MntABC proteins from *Escherichia coli*, *Staphylococcus aureus*, *Paracoccus denitrificans*, and *Streptococcus pneumonia* were obtained from GenBank resources (National Center for Biotechnology Information, https://www.ncbi.nlm.nih.gov/ (accessed on 10 October 2021)) for sequence alignment by using CLUSTALW software (http://www.genome.jp/tools-bin/clustalw (accessed on 18 October 2021)). Protein structure prediction was achieved by AlphaFold2 online tools (https://colab.research.google.com/github/deepmind/alphafold/blob/main/notebooks/AlphaFold.ipynb (accessed on 8 June 2024)). Structural representations were generated using PyMOL (http://www.pymol.org/ (accessedon 25 January 2022)) and ChimeraX (https://www.cgl.ucsf.edu/chimera/ (accessed on 21 March 2022)).

Molecular docking studies were conducted using the AutoDock Vina algorithm. Prior to docking, ligand and receptor molecules were preprocessed using AutoDock Tools [35]. A grid box of 40 × 40 × 40 Å with a spacing of 0.375 Å was defined to encompass the target binding site. The docking procedure employed the Lamarckian Genetic Algorithm (LGA) for conformational sampling and energy minimization. The resulting ligand poses were ranked according to their calculated binding energy (ΔG) values derived from the docking analysis.

### 4.3. Targeted Gene Disruption via Homologous Recombination

Single gene deletion mutant of the *dr2283*, *dr2284*, and *dr2523* (*Δdr2283*, *Δdr2284*, *Δdr2523*) was constructed through an optimized tripartite ligation strategy combined with homologous recombination, modified from previously described methods [36]. Primers (P1/P2: *Bam*HI site; P3/P4: *Hind*III site; P5/P6: internal validation) were designed via Primer 5.0 based on KEGG-derived sequences. All primers used in this study are listed in Appendix A. Upstream/downstream ~1 kbp fragments of targeted gene were amplified from *D. radiodurans* R1 genomic DNA and the kanamycin resistance cassette was amplified from pRADK plasmid. PCR products were digested with *Bam*HI/*Hind*III (37 °C, 5 h). Fragments were ligated using T4 DNA ligase (16 °C, overnight) and transformed into CaCl_2_-treated R1 competent cells. Transformants were incubated in TGY (30 °C, 20 h), plated on TGY-kanamycin agar plates (30 °C, 2–3 days), and validated by PCR and sequencing. Confirmed mutants were stored at −80 °C.

The *mntABC* deletion mutant was generated based on the *Δdr2523* strain using homologous recombination. Upstream (*Δdr2284*-P1/P2) and downstream (*Δdr2283*-P3/P4) homology fragments were amplified from *D. radiodurans* R1 genomic DNA, digested with *BamH*I/*Hind*III, and ligated to a streptomycin resistance cassette overnight at 16 °C. The ligated construct was transformed into *Δdr2523* competent cells. Transformants were selected on TGY agar plates supplemented with kanamycin and streptomycin (TKS), validated by PCR and sequencing, and designated *ΔdrMntABC*.

### 4.4. Gene Complementation

For gene complementation, the genes of *dr2283*, *dr2284*, and *dr2523* were amplified from wild-type R1 using primer pairs dr2283c-F/R, dr2284c-F/R, and dr2523c-F/R (Appendix A), respectively. These fragments were subsequently cloned into the pRADK vector to generate recombinant plasmids pRAD-dr2283, pRAD-dr2284, and pRAD-dr2523. Each plasmid was then transformed into its corresponding mutant strain, yielding the complemented strains *Δdr2283/c-dr2283*, *Δdr2284/c-dr2284*, and *Δdr2523/c-dr2523*. All mutant and complemented strains were validated through PCR amplification and sequencing.

### 4.5. Site-Directed Mutagenesis

Site-directed mutagenesis was performed using the QuickChange Kit (Stratagene, La Jolla, CA, USA). Mutated sequences were amplified from pRAD-dr2283, pRAD-dr2284, and pRAD-dr2523 with their mutagenesis primers (Appendix A), treated with *Dpn*I to remove methylated templates, and ligated into pRADK. The resultant plasmids were verified by sequencing and individually transformed into their corresponding mutant. All site-mutation complemented strains were validated via PCR and sequencing.

### 4.6. Subcellular Localization Analysis of DrMntABC

The DNA fragments of DrMntABC were fused to *eGFP* in plasmid pRADG under the control of the *groEL* promoter, generating pRADG-dr2283, pRADG-dr2284, and pRADG-dr2523. The constructs and empty vector (pRADG control) were transformed into wild-type *D. radiodurans* R1. Transformants were selected via chloramphenicol resistance and verified by sequencing.

Exponential-phase cells (OD_600_ 1.0) were immobilized on glass slides and imaged using a Zeiss LSM510 confocal microscope(Zeiss, Oberkochen, Germany). Nucleoids were counterstained with DAPI (blue fluorescence). Spatial location of the eGFP-MntABC signals and DAPI-stained nucleoids was analyzed using ImageJ 1.47 (NIH, New York, NY, USA).

### 4.7. Metal Cation Sensitivity Assessment

The metal ion sensitivity assay was performed to evaluate the role of the MntABC transporter in metal homeostasis, using the method described by Rosch et al. [30]. Aqueous stock solutions (1 M) of divalent metal chlorides—MnCl_2_, CdCl_2_, CrCl_2_, ZnCl_2_, CuCl_2_, and FeCl_2_ (Sigma-Aldrich, St. Louis, MO, USA)—were prepared in ultrapure Milli-Q water and sterilized via membrane filtration (0.22 μm pore size). Wild-type *D. radiodurans* R1 and mutant strains were cultured to exponential-phase cells (OD_600_ 1.0), and 300 μL of fresh culture was spread evenly onto TGY agar plates. Sterile filter disks (5 mm diameter) were aseptically placed at the center of each plate. Aliquots (5 μL) of each metal ion solution were pipetted onto the disks. Plates were incubated upright at 30 °C until the liquid was absorbed, then inverted and cultured for 1–2 days. Inhibition zones were imaged, and the distance between the disk edge and bacterial growth was measured to quantify metal sensitivity.

### 4.8. Intracellular Metal Ion Content Analysis

Intracellular metal ion content was analyzed through inductively coupled plasma mass spectrometry (ICP-MS) [30]. *D. radiodurans* strains were cultured in 500 mL Chelex-100-treated TGY broth to eliminate background metal contamination, subsequently supplemented with a defined metal cocktail (50 μM each of MnCl_2_, ZnCl_2_, CuCl_2_, and FeCl_2_). Exponential-phase cells (OD_600_ 1.0) were pelleted via centrifugation (10,000× *g*, 10 min, 4 °C) and subjected to sequential washing: three cycles with PBS-EDTA (1 mM EDTA, pH 7.5) to remove surface-bound cations, followed by three PBS rinses to eliminate chelator residues. Lyophilized biomass was digested in 5 mL ultrapure 65% nitric acid (Sigma-Aldrich, USA) with 1 mL 30% H_2_O_2_ at 100 °C for 2 h using a PTFE-lined digestion vessel.

Quantification of metal ion concentration was performed on an ELAN DRC-e ICP-MS system (PerkinElmer, Waltham, MA, USA), calibrated with NIST-traceable multi-element standards. Non-metal-supplemented cultures processed identically served as background controls. Ion concentrations were normalized to cellular dry weight determined gravimetrically. Triplicate biological replicates were analyzed, with data expressed as mean ± SD. 

### 4.9. Determination of Bacterial Growth Curves

Single colonies of the wild-type strain and the constructed mutant strains were inoculated into 5 mL of fresh TGY liquid medium (supplemented with appropriate antibiotics for mutant strains) and cultured to the exponential-phase cells (OD_600_ 1.0). Aliquots (200 μL) of each culture were transferred into 10 mL of fresh TGY liquid medium and grown to an OD_600_ of 0.1. Each group was prepared in triplicate and incubated at 30 °C with shaking (200 rpm). OD_600_ values were measured every 2 h over a 28 h period using a spectrophotometer, with sterile TGY medium serving as the blank for instrument calibration. Growth curves were plotted with time (h).

### 4.10. Real-Time Quantitative PCR Analysis

The gene expression level under various stress conditions was analyzed by real-time quantitative PCR (qRT-PCR). Bacterial cultures were grown to an OD_600_ of 0.6 and were exposed to oxidative stress (30–60 mM H_2_O_2_) or γ-irradiation treatment (kGy) for 1 h. Cells were pelleted via centrifugation (5000× *g*, 4 °C), followed by total RNA extraction from 5 mL cultures utilizing TRIZOL reagent (Invitrogen, Carlsbad, CA, USA). cDNA synthesis was conducted in 20 μL reaction volumes containing 1 μg DNase I-treated RNA and 3 μg random hexamers. qRT-PCR amplification employed SYBR Premix Ex Taq (TaKaRa, Kusatsu, Japan) with primer pairs detailed in Appendix A. Relative transcript levels were quantified using the 2^−ΔΔCt^ method, with *dr1343* (encoding glyceraldehyde-3-phosphate dehydrogenase, GAPDH) serving as the endogenous control. All reactions were performed on the Stratagene Mx3005P real-time PCR detection system(Agilent, Santa Clara, CA, USA).

### 4.11. Survival Analysis

Cells grown in TGY broth to OD_600_ 0f 1.0 were pelleted and resuspended in PBS. A 100 μL sample of cell suspension was diluted with PBS to 10^7^ colony-forming units (CFU) mL^−1^. For oxidative stress, suspensions were exposed to gradient concentrations of H_2_O_2_ for 30 min, followed by plating on TGY agar and colony enumeration after 3 days. For γ-ray irradiation assays, cell suspensions were treated with ^60^Co γ-ray for 1 h on ice, with radiation doses controlled by modulating the exposure distance of samples from the γ-ray source. Following the treatment, cells were plated on TGY agar, incubated at 30 °C for 3 days, and viable colonies were quantified. Data are presented as mean ± standard deviation (SD) from triplicate biological replicates.

### 4.12. Intracellular ROS Accumulation Assay

ROS accumulation in *D. radiodurans* strains in the presence or absence of 60 mM H_2_O_2_ treatment were analyzed using the method of 2′2′,7′7′-dichlorofluorescein diacetate (DCFH-DA) assay, as described previously [31]. Briefly, 2 mL aliquots of cell cultures (OD_600_ = 0.8) underwent three washes with phosphate-buffered saline (PBS). The resulting pellets were reconstituted in DCFH-DA solution and incubated at 37 °C for 30 min. Post-incubation, cells were washed thrice with PBS and finally suspended in 2 mL PBS. Subsequently, 1 mL portions of this suspension were exposed to either 50 mM H_2_O_2_ or left untreated for 30 min. Within cells, cellular esterases hydrolyze DCFH-DA to DCFH, which is subsequently oxidized by ROS to the fluorescent compound dichlorofluorescein (DCF). DCF fluorescence intensity was measured using SpectraMax M5 fluorescence spectrophotometer (Molecular Devices, Sunnyvale, CA, USA) (excitation: 485 nm, emission: 525 nm).

### 4.13. Yeast Two-Hybrid Assay

The yeast two-hybrid assay was conducted by cloning the *dr2283*, *dr2284*, *dr2523*, and their site-mutation gene fragments into the pGBKT7 (bait vector) and pGADT7 (prey vector) plasmids, respectively. Interaction between the expressed bait and prey proteins activates transcription of the *lacZ* and *his3* reporter genes, enabling yeast growth on quadruple dropout medium (SD/-Ade/-His/-Leu/-Trp). For competent cell preparation, yeast strain AH109 was streaked onto YPDA solid medium and incubated at 30 °C overnight. A single colony was inoculated into 5 mL YPDA liquid medium and cultured to saturation. Subsequently, 5 mL of the saturated culture was transferred to 100 mL YPDA liquid medium and grown to an OD_600_ of ~0.5. Cells were harvested by centrifugation at 3000× *g* for 5 min at room temperature (RT), resuspended in 20 mL sterile ddH_2_O, and centrifuged again. The pellet was resuspended in 10 mL sterile 1× TE/LiAc solution, centrifuged, and finally resuspended in 500 μL sterile 1× TE/LiAc solution, followed by a 10 min incubation at RT. Plasmids were transformed into the competent yeast cells via the lithium acetate method, and transformants were initially selected on double dropout medium (SD/-Leu/-Trp). Positive colonies were subsequently streaked onto quadruple dropout medium (SD/-Ade/-His/-Leu/-Trp) to confirm protein–protein interactions.

### 4.14. Statistical Analysis

Student’s *t*-test was used to assess the significance between results, and *p* < 0.05 was considered as significant.

## 5. Conclusions

In conclusion, the MntABC system is a cornerstone of manganese homeostasis and oxidative stress resistance in *Deinococcus radiodurans*. By facilitating manganese ion uptake, MntABC shields cellular proteins from oxidative damage, thereby enhancing cellular viability. The functional integrity of this transporter, which is essential for its Mn^2+^ transport activity, relies on critical interactions between its subunits. Our findings not only elucidate the mechanism by which MntABC maintains a high intracellular Mn/Fe ratio to bolster antioxidant defenses but also significantly advance the broader understanding of manganese transport systems across the bacterial domain.

## Figures and Tables

**Figure 1 ijms-26-09407-f001:**
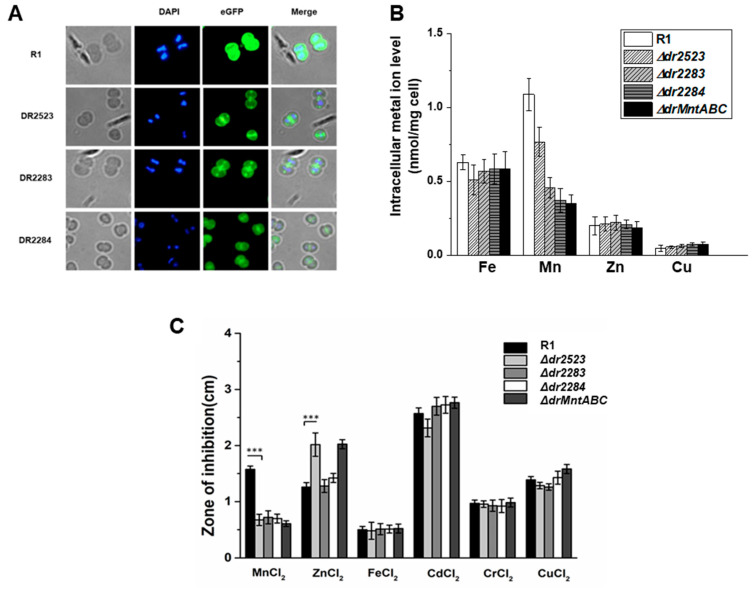
The MntABC homologs jointly form an Mn transporter system in *D. radiodurans*. (**A**) Fluorescence labeling analysis of the localization of the DR2523, DR2283, and DR2284 in cells. eGFP was fused to the N-terminus of each protein and expressed in the wild-type strain R1. *D. radiodurans* R1 strain transformed with an empty pRADG vector containing the eGFP gene was used as a control, in which the eGFP was localized in the cytoplasm. The nucleoid was stained with DAPI (blue fluorescence). The merged image indicated the co-localization of the eGFP-labeled protein and cell membrane. (**B**) Intracellular concentration of metal ions in the wild type and mutants cultured in TGY medium. (**C**) Metal ion sensitivity assays. Wild-type R1 and deletion mutants were cultured on TGY plates overlaid with filter disks saturated with 1 M solutions of various cations. The zone of inhibition was measured from the edge of the disk after three days. R1, the wild type; *Δdr2523*, *dr2523* mutant; *Δdr2283*, *dr2283* mutant; *Δdr2284*, *dr2284* mutant; *ΔdrMntABC*, MntABC mutant. ***, *p* < 0.001.

**Figure 2 ijms-26-09407-f002:**
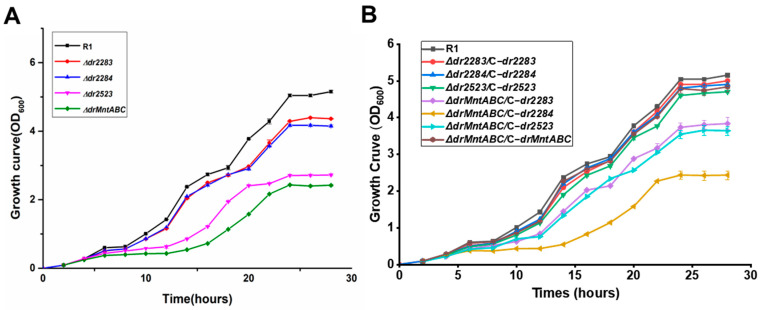
Deficiency of the MntABC impacts on cell growth. (**A**) Growth curve of *D. radiodurans* wild-type strain R1 and MntABC mutant strains. (**B**) Growth curve of *D. radiodurans* MntABC mutant strains complemented by the corresponding genes.

**Figure 3 ijms-26-09407-f003:**
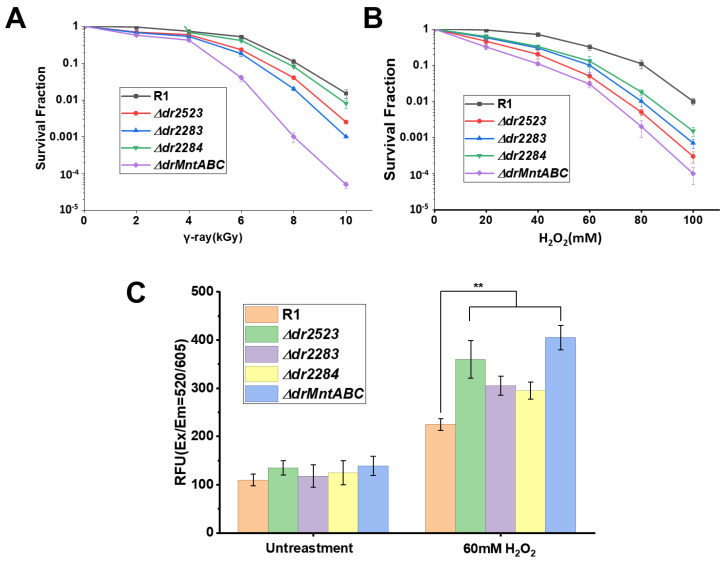
MntABC is involved in resistance of *D. radiodurans* to oxidative stress. (A,B) Survival curves of *D. radiodurans* strains following exposure to γ-ray radiation (**A**) and H_2_O_2_ (**B**), respectively. R1, wild-type strain; (**C**) ROS accumulation in *D. radiodurans* strains in the presence or absence of 60 mM H_2_O_2_ treatment. **, *p* < 0.01.

**Figure 4 ijms-26-09407-f004:**
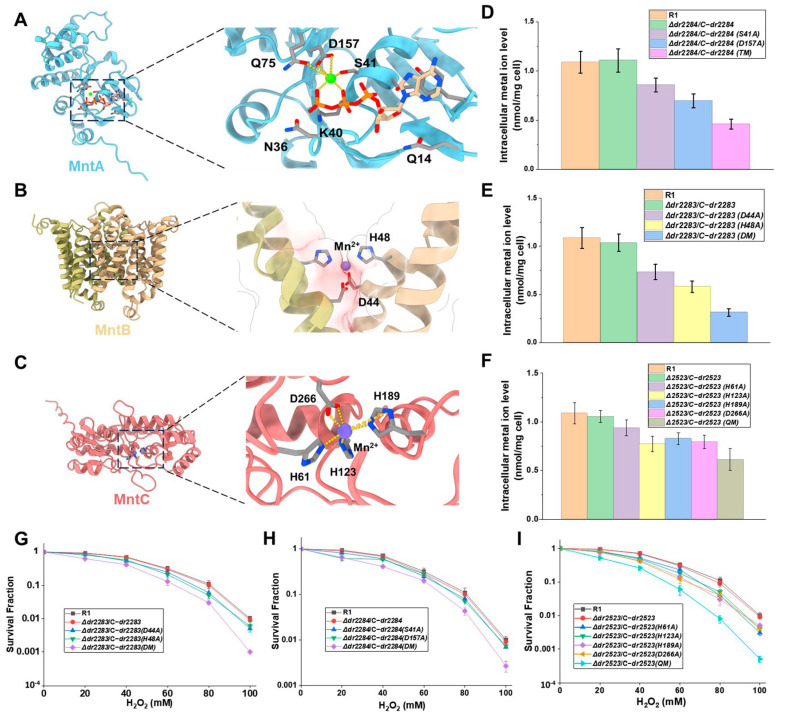
The key sites within MntABC proteins that are involved in Mn ion transport and oxidative resistance. (**A**–**C**) Protein structure modeling and key sites analysis of MntA (**A**), MntB (**B**), and MntC (**C**). Each protein structure was predicted by AlphaFold2 and the substrate was docked to protein by using AutoDock Vina. Green, Mg ions; Stick, ATP. (**D**–**F**) Intracellular concentration of Mn ions in the *D. radiodurans* wild-type strain, and mutants complemented with the corresponding site-mutation gene. Cells were cultured in TGY medium. (**G**–**I**) Survival curves of *D. radiodurans* wild-type strain, and mutant strains that are complemented with the corresponding site-mutation gene following exposure to H_2_O_2_.

**Figure 5 ijms-26-09407-f005:**
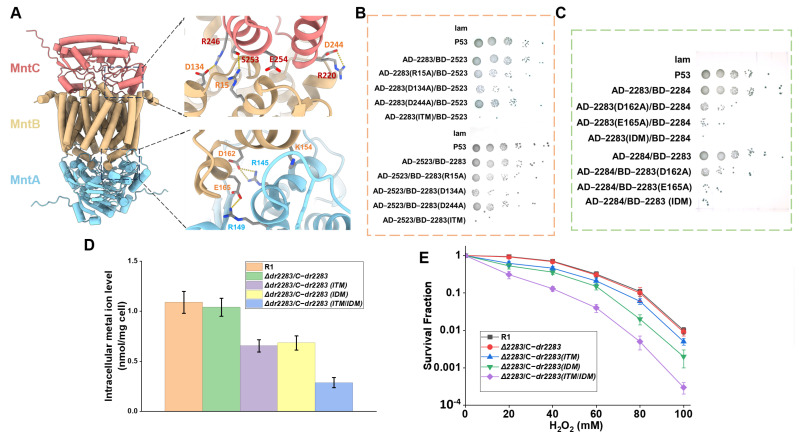
Effect of interactions between proteins in the MntABC system on Mn ion transport and oxidative stress resistance of *D*. *radiodurans*. (**A**) Structural modeling and site analysis of the MntABC complexes using AlphaFold2. The potential interacting residues were indicated in the enlarged image. (**B**) Yeast two-hybrid analysis of the interaction between DrMntB and DrMntC. (**C**) Yeast two-hybrid analysis of the interaction between DrMntB and DrMntA. (**D**) Intracellular concentration of Mn ions in the wild type R1, and *drmntB* (*dr2283*) mutant complemented with site-mutation *drmntB* gene. Cells were cultured in TGY medium. (**E**) Survival curves of *D. radiodurans* wild-type strain R1, and *drmntB* (*dr2283*) mutant complemented with site-mutation *drmntB* gene following exposure to H_2_O_2_.

## Data Availability

The raw data supporting the conclusions of this article will be made available by the authors without undue reservation.

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
