# Peer review of "The Role of the MntABC Transporter System in the Oxidative Stress Resistance of Deinococcus radiodurans"

_ijms, 2025, doi:10.3390/ijms26199407_

Round 1
Reviewer 1 Report
Comments and Suggestions for Authors
This manuscript investigates the role of the MntABC transporter in manganese uptake and oxidative stress tolerance in Deinococcus radiodurans. The authors combine fluorescence-based localization, targeted mutagenesis, stress (Hâ‚‚Oâ‚‚ and γ-irradiation), and structure/docking analyses to present a coherent case that MntABC is central to Mn homeostasis and stress resistance.
Some comments and questions:
1. Please revise for stylistic precision and spacing errors that occasionally obscure meaning (e.g., lines 65, 104–105, 293). Standardize gene/protein nomenclature throughout. Instances such as “dr2523” vs “dr_2523” (e.g., lines 78, 80 and elsewhere).
2. Figure 1B - The x-axis description is missing.
3. Lines 150–151 - For clarity, refer to strains lacking a gene as “deletion mutants” or “knockout mutants,” especially since later sections introduce amino-acid-substitution mutants. Consistent terminology will help readers distinguish the variants.
4. Lines 423, 433 - Please specify the exact concentrations of nitric acid and Hâ‚‚Oâ‚‚ used.
5. Lines 408, 418, 429 in methods - use a single, consistent term for exponential growth phase instead of three.
6. Lines 325,326 - Please clarify the logic here. Did the authors mean that they used PsaA as a reference structure for prediction?
7. Please explain why the reporter fluorescent protein was fused at the N-terminus of the target protein. For proteins expected to be exported to the bacterial periplasm, N-terminal signal peptides are often required; an N-terminal reporter can disrupt export/processing. If periplasmic localization is intended, clarify how this was accommodated or provide a rationale for N-terminal tagging versus C-terminal fusions. Line 404: the plasmid names likely should reflect the fusion, e.g., pRADG-eGFP-dr2283 (and others). Please also provide the N-terminal sequence context (tag order, linker length/sequence, presence of signal peptide, any motifs).
Author Response
Comments 1: Please revise for stylistic precision and spacing errors that occasionally obscure meaning (e.g., lines 65, 104–105, 293). Standardize gene/protein nomenclature throughout. Instances such as “dr2523” vs “dr_2523” (e.g., lines 78, 80 and elsewhere).
Response 1: Thank you for your helpful comment and suggestion. All instances of gene/protein nomenclature have been standardized throughout the manuscript, and the underscores (e.g., in “dr_2523”) have been removed for consistency.
Comments 2: Figure 1B - The x-axis description is missing.
Response 2: We appreciate this comment. The x-axis label in Figure 1B has now been included.
Comments 3: Lines 150–151 - For clarity, refer to strains lacking a gene as “deletion mutants” or “knockout mutants,” especially since later sections introduce amino-acid-substitution mutants. Consistent terminology will help readers distinguish the variants.
Response 3: We agree with this suggestion. The term “deletion mutants” has been consistently applied to refer to strains lacking the gene, in order to clearly distinguish them from amino-acid-substitution mutants introduced later.
Comments 4: Lines 423, 433 - Please specify the exact concentrations of nitric acid and Hâ‚‚Oâ‚‚ used.
Response 4: Thank you for your helpful comment. The specific concentrations of nitric acid and Hâ‚‚Oâ‚‚ have now been provided in the respective sections.
Comments 5: Lines 408, 418, 429 in methods - use a single, consistent term for exponential growth phase instead of three.
Response 5: We thank the reviewer for this comment. The terminology has been unified to “exponential-phase cells” throughout the Methods section.
Comments 6: Lines 325,326 - Please clarify the logic here. Did the authors mean that they used PsaA as a reference structure for prediction?
Response 6: We appreciate the opportunity to clarify. We meant that although the homologous protein PsaA has been associated with similar functions, it was not used as a template for structural prediction in this study.
Comments 7: Please explain why the reporter fluorescent protein was fused at the N-terminus of the target protein. For proteins expected to be exported to the bacterial periplasm, N-terminal signal peptides are often required; an N-terminal reporter can disrupt export/processing. If periplasmic localization is intended, clarify how this was accommodated or provide a rationale for N-terminal tagging versus C-terminal fusions. Line 404: the plasmid names likely should reflect the fusion, e.g., pRADG-eGFP-dr2283 (and others). Please also provide the N-terminal sequence context (tag order, linker length/sequence, presence of signal peptide, any motifs).
Response 7: We thank the reviewer for raising these important points. Regarding the N-terminal fusion, we predicted the signal peptides of MntABC proteins using SignalP and found no significant signal peptide domains, suggesting that N-terminal tagging should not interfere with its sublocalization. Furthermore, we attempted C-terminal GFP fusions but observed suppressed fluorescence signals, likely due to impaired protein folding or activity. Therefore, the N-terminal fusion strategy was adopted. The plasmid pRADG denotes the pRAD-eGFP vector, a standard fluorescent expression plasmid used in Deinococcus radiodurans, as described in our previous work (doi: 10.1371/journal.pone.0202287). The pRADG vector includes a flexible linker between eGFP and the target protein.
Reviewer 2 Report
Comments and Suggestions for Authors
Review of the paper by Binqiang Wang et. Al.
Deinococcus radiodurans is known for its phenomenal radiation resistance along with its resistance to dehydration and other extreme environmental conditions. Molecular mechanisms that confer such resistances, and the genes that enable such behavior is interesting to study. In this work, Wang et Al. took effort to decipher molecular mechanisms and gene functions responsible for Mn ion homeostasis. In Deinococcus radiodurans, Mn ions forms complex with small molecules and such complexes are proposed to confer radiation resistance. Mn ion is imported in the cells by MntABC transporter system, and in this work, authors analyzed the membrane localization of this transporter system and characterized the effect of certain mutations on the efficiency of Mn import and its underlaying role in cellular growth and radiation resistance.
Authors addressed an interesting question in this work, overall, the study is well designed, and claims are mostly justified. We would like to recommend the publication of this work with minor modifications as suggested below.
Minor comments:
- Please consider adding a schematic drawing of the cell, schematically showing the sub cellular localization of different transporter proteins, enzymes involved in this process.
- Please mention the full description of TGY media, instead of the abbreviation for more generalized reader, when the term is first introduced.
- In figure 1B, the Y axis is mentioned as intracellular metal ion level, in the text authors specified this is intracellular Mn level, which should be mentioned clearly on the axis. Also, the unit is nmol/mg of the cell, please mention clearly if the unit ‘mg’ representing cellular dry weight, we found this analysis confusing, and authors should take effort to elaborate on this.
- Please show the data that other intracellular metal ion concentrations (Fe, Cu) are not changed in the nutant. We could not find this data; this is crucial to show to claim the MntABC is specific.
- Please show the plate images for the zone of inhibition experiments in a supplementary figure.
- In figure 2, please show the growth curve where the OD is plotted in Log scale, in linear scale this is difficult to compare the slopes of different mutants. This should be shown in log scale, also consider including a table mentioning doubling time and growth rates. This is important for the reader to compare mutants.
- Please consider expanding the methods section more extensively.
Author Response
Comments 1: Please consider adding a schematic drawing of the cell, schematically showing the subcellular localization of different transporter proteins, enzymes involved in this process.
Response 1: We thank the reviewer for this suggestion. We agree that a schematic diagram would help illustrate the subcellular localization of the relevant transporters and enzymes. However, as the main focus of our study is on functional characterization rather than establishing new localization patterns, and because the localization of metal ion channels has been well-documented in previous reviews (e.g., https://doi.org/10.1016/j.resmic.2019.11.002), we have opted not to include a schematic in the current manuscript. We will consider incorporating such illustrations in future studies to provide a more comprehensive overview.
Comments 2: Please mention the full description of TGY media, instead of the abbreviation for more generalized reader, when the term is first introduced.
Response 2: We thank the reviewer for this comment. We have now provided the full description of TGY medium (containing tryptone, glucose, and yeast extract) at its first mention in the text.
Comments 3: In figure 1B, the Y axis is mentioned as intracellular metal ion level, in the text authors specified this is intracellular Mn level, which should be mentioned clearly on the axis. Also, the unit is nmol/mg of the cell, please mention clearly if the unit ‘mg’ representing cellular dry weight, we found this analysis confusing, and authors should take effort to elaborate on this.
Response 3: We thank the reviewer for this comment. In the text, we described the intracellular metal ion level of several metal ions, and discussed specially intracellular Mn level. Regarding the unit, "nmol/mg" refers to nmol per milligram of cellular dry weight, as consistently used in our previous studies (DOI: 10.1371/journal.pone.0202287; DOI: 10.1186/1471-2180-10-319; DOI: 10.1371/journal.pone.0106341).
Comments 4: Please show the data that other intracellular metal ion concentrations (Fe, Cu) are not changed in the nutant. We could not find this data; this is crucial to show to claim the MntABC is specific.
Response 4: We sincerely apologize for the confusion caused by the incomplete figure X axis. The data for Fe and Cu were indeed included in the original submission but were not clearly visible due to the lackness of an axis labeling. We have corrected Figure 1B, and the data for these metals are now properly displayed, supporting the specificity of MntABC for manganese.
Comments 5: Please show the plate images for the zone of inhibition experiments in a supplementary figure.
Response 5: We thank the reviewer for this suggestion. Due to the large number of metal types and bacterial strains tested, presenting all plate images clearly and legibly in the main text or supplementary materials was challenging. To ensure clarity and highlight the most relevant findings, we have selected and included representative plate images showing the zones of inhibition for Mn/Zn treatments, which exhibit the most significant differences, as new Supplementary Figure S5.
Comments 6: In figure 2, please show the growth curve where the OD is plotted in Log scale, in linear scale this is difficult to compare the slopes of different mutants. This should be shown in log scale, also consider including a table mentioning doubling time and growth rates. This is important for the reader to compare mutants.
Response 6: We appreciate the reviewer’s feedback. However, as the growth phenotypes were highly reproducible across three independent biological replicates and the differences between strains are clearly observable in the linear-scale plot, we believe that the current presentation sufficiently supports our conclusions. Therefore, we prefer to retain the original format for Figure 2. We hope the reviewer agrees that the existing data clearly demonstrate the growth trends we describe.
Comments 7: Please consider expanding the methods section more extensively.
Response 7: We thank the reviewer for this comment. We have now carefully reviewed and expanded the Methods section to provide additional details where necessary, ensuring that all experimental procedures are described with sufficient clarity and reproducibility.